# Unique miRomics Expression Profiles in *Tannerella forsythia*-Infected Mandibles during Periodontitis Using Machine Learning

**DOI:** 10.3390/ijms242216393

**Published:** 2023-11-16

**Authors:** Chairmandurai Aravindraja, Syam Jeepipalli, William Duncan, Krishna Mukesh Vekariya, Sakshee Bahadekar, Edward K. L. Chan, Lakshmyya Kesavalu

**Affiliations:** 1Department of Periodontology, College of Dentistry, University of Florida, Gainesville, FL 32610, USA; aravindrchairman@ufl.edu (C.A.); sjeepipalli@dental.ufl.edu (S.J.); kvekariya@ufl.edu (K.M.V.); 2Department of Community Dentistry, College of Dentistry, University of Florida, Gainesville, FL 32610, USA; duncanw@ufl.edu; 3Department of Computer and Information Science and Engineering, University of Florida, Gainesville, FL 32610, USA; sbahadekar@ufl.edu; 4Department of Oral Biology, College of Dentistry, University of Florida, Gainesville, FL 32610, USA; echan@dental.ufl.edu

**Keywords:** periodontal disease, miRNAs, NanoString analysis, transient miRNA expression, machine learning, *T. forsythia*

## Abstract

*T. forsythia* is a subgingival periodontal bacterium constituting the subgingival pathogenic polymicrobial milieu during periodontitis (PD). miRNAs play a pivotal role in maintaining periodontal tissue homeostasis at the transcriptional, post-transcriptional, and epigenetic levels. The aim of this study was to characterize the global microRNAs (miRNA, miR) expression kinetics in 8- and 16-week-old *T. forsythia*-infected C57BL/6J mouse mandibles and to identify the miRNA bacterial biomarkers of disease process at specific time points. We examined the differential expression (DE) of miRNAs in mouse mandibles (*n* = 10) using high-throughput NanoString nCounter^®^ miRNA expression panels, which provided significant advantages over specific candidate miRNA or pathway analyses. All the *T. forsythia*-infected mice at two specific time points showed bacterial colonization (100%) in the gingival surface, along with a significant increase in alveolar bone resorption (ABR) (*p* < 0.0001). We performed a NanoString analysis of specific miRNA signatures, miRNA target pathways, and gene network analysis. A total of 115 miRNAs were DE in the mandible tissue during 8 and 16 weeks The *T. forsythia* infection, compared with sham infection, and the majority (99) of DE miRNAs were downregulated. nCounter miRNA expression kinetics identified 67 downregulated miRNAs (e.g., miR-375, miR-200c, miR-200b, miR-34b-5p, miR-141) during an 8-week infection, whereas 16 upregulated miRNAs (e.g., miR-1902, miR-let-7c, miR-146a) and 32 downregulated miRNAs (e.g., miR-2135, miR-720, miR-376c) were identified during a 16-week infection. Two miRNAs, miR-375 and miR-200c, were highly downregulated with >twofold change during an 8-week infection. Six miRNAs in the 8-week infection (miR-200b, miR-141, miR-205, miR-423-3p, miR-141-3p, miR-34a-5p) and two miRNAs in the 16-week infection (miR-27a-3p, miR-15a-5p) that were downregulated have also been reported in the gingival tissue and saliva of periodontitis patients. This preclinical in vivo study identified *T. forsythia*-specific miRNAs (miR-let-7c, miR-210, miR-146a, miR-423-5p, miR-24, miR-218, miR-26b, miR-23a-3p) and these miRs have also been reported in the gingival tissues and saliva of periodontitis patients. Further, several DE miRNAs that are significantly upregulated (e.g., miR-101b, miR-218, miR-127, miR-24) are also associated with many systemic diseases such as atherosclerosis, Alzheimer’s disease, rheumatoid arthritis, osteoarthritis, diabetes, obesity, and several cancers. In addition to DE analysis, we utilized the XGBoost (eXtreme Gradient boost) and Random Forest machine learning (ML) algorithms to assess the impact that the number of miRNA copies has on predicting whether a mouse is infected. XGBoost found that miR-339-5p was most predictive for mice infection at 16 weeks. miR-592-5p was most predictive for mice infection at 8 weeks and also when the 8-week and 16-week results were grouped together. Random Forest predicted miR-592 as most predictive at 8 weeks as well as the combined 8-week and 16-week results, but miR-423-5p was most predictive at 16 weeks. In conclusion, the expression levels of miR-375 and miR-200c family differed significantly during disease process, and these miRNAs establishes a link between *T. forsythia* and development of periodontitis genesis, offering new insights regarding the pathobiology of this bacterium.

## 1. Introduction

Over the past decade, numerous studies have reported that non-coding RNAs (ncRNAs) including microRNAs (miRNAs), circular RNAs (circRNAs), transfer RNAs (60–95 nucleotides (nt); tRNA), transfer RNA-derived small RNAs (15–45 nt; tsRNA), and long non-coding RNAs (lncRNAs) are important epigenetic regulatory mechanisms. miRNAs are 21–25 nt contained, small, non-coding, regulatory RNAs that negatively regulate gene expression by directly binding to the 3′ UTR regions of mRNAs [1,2,3,4]. Based on the sequence complementarity, these miRNAs control the translational regulation of the mRNAs or lead to their degradation pathways. Currently, more than 2000 miRNAs are available in the public database and each specific miRNA can have hundreds of different targets that affect the majority of gene expression networks [5,6]. miRNAs are implicated in a wide spectrum of infectious diseases, various cancers, Alzheimer’s disease, diabetes, autoimmune diseases, and numerous biological functions (cell differentiation, development, proliferation, stress response and apoptosis). miRNAs are able to resist degradation by ribonucleases that have the potential to be biomarkers for disease prediction and diagnosis. Accordingly, an improved understanding of the biogenesis pattern of miRNAs could potentially lead to the development of novel diagnostic biomarkers for polymicrobial infection-driven inflammatory diseases. Periodontal disease (PD) is a dysbiotic chronic inflammatory disease caused by microbes interacting in the human subgingival sulcus [7,8]. Interaction of various microbes in the human subgingival sulcus results in host immune and inflammatory responses, leading to alveolar bone resorption and subsequent tooth loss. Remarkably, little is known about the effect of periodontal bacteria on host miRNA expression and the role of host miRNAs on modulating bacterial infections. *T. forsythia* is one of the red complex bacteria in the subgingival cavity most frequently described as a cause of gingival infection and PD [9]. The virulence factors from *T. forsythia* (surface antigen BspA, cell surface proteolytic enzymes, hemagglutinin, cell envelope lipoproteins, glycosidases, cell surface (S)-layer) can disrupt the epithelial cells [10] and stimulate host immune responses [11]. After its stimulation, *T. forsythia* can successfully evade the host immune response, enter the circulatory system and colonize distal organs such as the heart, liver, lungs, and aorta with significant CD3+ T cells and IL4R-positive CD3+ Th2 T cells in intimal and adventitial layers of aortas [11]. Our laboratory reported the intracellular localization of *T. forsythia* DNA in the aortic tissues and identified inflammatory mediators in the serum of the mice after gingival infection of *T. forsythia*. In vitro, *T. forsythia* has been shown to co-aggregate and form synergistic biofilm with *F. nucleatum* [12]. Ancient skeletal remains of Wichita Ancestors from 20 archeological sites (dating approximately to 1250-1450 CE) showed a high abundance of bacteria typically associated with periodontitis such as *T. forsythia* and *Treponema denticola* [13].

Several inflammatory miRNAs such as miR-21 [14], miR-146a [15,16,17] miR-126-3p [16,18,19] have been reported to be involved in the initiation and progression of PD. However, none of these inflammatory miRNAs were differentially expressed in our recent report that used partial human mouth microbes (PAHMM) in an ecological time-sequential polybacterial infection mouse model (ETSPPI) [20]. In contrast, five miRs (miR-195, 31, 125b-5p, 15a, 423-5p) that were differentially expressed during the *Porphyromonas gingivalis* monobacterial infection are also expressed in chronic periodontitis [21]. Similarly, seven miRs (miR-22, 486, 126-3p, 378, 151a-3p, 423-5p, 221) that were differentially expressed during *Treponema denticola* monobacterial infection are also expressed in chronic periodontitis with diabetes [21,22]. This observation suggested that several miRNAs are involved in PD and the miRNA biogenesis pattern may be bacterial specific and time dependent. As of now, there are no reports on the miRNA profiling during the progression of PD using *T. forsythia*, a major red complex bacterium strongly associated with both PD and peri-implantitis. Hence, in order to delineate the differential miRNA expression with specific bacterial infection leading to PD, this study utilized *T. forsythia* in the induction of experimental periodontitis at two different time points.

Based on our recent literature search, there are no studies that used live *T. forsythia* to analyze miRNA signature patterns under in vitro and in vivo conditions. Hence, studies showing the specific miRNA signature patterns against *T. forsythia* infection are warranted to understand the complex inflammatory pathways mediated by *T. forsythia*. miRNAs expression in our monobacterial infections have certain similarity and distinctive expressions. Furthermore, we applied the XGBoost and Random Forest machine learning (ML) algorithms to the NanoString data. Since no prior studies have utilized ML for periodontal miRNA analysis in the rodent model, we hypothesized that we could gain insight into which miRNAs were associated with periodontal disease infection.

## 2. Results

### 2.1. Chronic Infection of T. forsythia Effectively Colonized in Mice Gingival Surface

After the first two weeks of *T. forsythia* infection, gingival plaque samples from mice were taken for 16S rRNA gene specific colony PCR—50% of the mice in Group I and 10% of the mice in Group III tested positive for the *T. forsythia* gDNA. After 4 weeks of infection, 80% of the mice in Group I and 70% of the mice in Group III tested positive for the *T. forsythia* gDNA. Further, after 12 weeks of infection, all mice (100%) in Group III were shown positive for the presence of the *T. forsythia* 16S rRNA gene (Table 1). None of the sham-infected mice (Groups II and IV) were positive at any point for genomic DNA of *T. forsythia*. These results confirmed the colonization of the *T. forsythia* in both 8-week and 16-week *T. forsythia*-infected mice.

### 2.2. Higher Alveolar Bone Resorption (ABR) and Bacterial Dissemination to Distal Organs

Horizontal ABR measurements were conducted to identify the periodontal disease outcome after *T. forsythia* bacterial infection. Mice at both 8 weeks and 16 weeks of *T. forsythia* infection showed significantly higher ABR in the mandible (lingual) (adjusted *p*-value *=* 0.0002 for the 8-week group; *p* < 0.01 for the 16-week group) (Figure 1B,C). Similarly, a significantly higher (*p <* 0.0001) ABR was observed in maxilla palatal for 16-week *T. forsythia*-infected mice. No IgG antibody response was observed against *T. forsythia* in both 8-weeek and 16-week infected mice. None of the mice infected with *T. forsythia* showed bacterial dissemination to other distal organs such as heart, lungs, brain, liver, kidney and spleen.

### 2.3. NanoString Analysis of miRNAs in T. forsythia-Infected Mandibles

Global miRNA analysis was performed in the mandibles infected with *T. forsythia* at 8-week and 16-week infections. A *p*-value of >0.05 and fold change of 1.1 and above was considered for analysis and deemed to be significant (Table 2). nCounter miRNA expression profiling showed 67 downregulated miRNAs (e.g., miR-375, miR-200c, miR-200b, miR-34b-5p, miR-141) (Appendix A) in *T. forsythia*-infected mandibles compared to sham-infected mandibles during 8 weeks of infection. In total, 16 miRNAs were upregulated (e.g., miR-1902, miR-let-7c, miR-146a) (Table 3) and 32 miRNAs were found to be downregulated (e.g., miR-2135, miR-720, miR-376c) in *T. forsythia*-infected mice compared to sham-infected mice during the 16-week infection (Appendix A).

The volcano plot analysis identified eight downregulated (red) miRNAs and five upregulated (green) miRNAs that showed a fold difference of +1.1 with *p*-value of <0.05 in 8 week *T. forsythia*-infected mice compared to 16-week-infected mice (Figure 2A). The log2 fold change is on the *x* axis, and the negative log of the *p*-value is on the *y* axis. The black dots represent the miRNAs that do not pass the filter parameters. Surprisingly, no miRs showed a higher expression during 8 weeks of infection. In contrast, 16 miRs (e.g., miR-1902, miR-let-7c, miR-146a, miR-423-5p) were upregulated during 16 weeks of infection with *T. forsythia*-infected mandibles (Figure 2B and Table 3). Among the upregulated miRs, seven miRNAs (miR-423-5p, miR-210, miR-146a, miR-let-7a, miR-24, miR-218, miR-24b) are associated with chronic periodontitis (Table 3). Previously identified inflammatory miR-146a was DE during the *T. forsythia* infection. Further, several DE miRNAs that were significantly upregulated (e.g., miR101b, miR-218, miR-127, miR-24) are associated with many systemic diseases such as atherosclerosis, Alzheimer’s disease, rheumatoid arthritis, osteoarthritis, diabetes, obesity, and several cancers (Table 3). Detailed reported functions of the top five downregulated miRNAs (miR-375, miR-200c, miR-200b, miR-141, miR-34b-5p) during an 8-week infection with *T. forsythia* are shown in Table 4. Among the 67 downregulated DE miRNAs, 2 miRNAs miR-375 (−2.38 FC) and miR-200c (−2.6 FC) were downregulated <2.6-fold during an 8-week *T. forsythia* infection (Table 4).

### 2.4. Functional Pathways Analysis of DE miRNAs

Predicted functional pathway analysis of the DE miRNAs in both 8-week and 16-week *T. forsythia*-infected mice identified several pathways (DIANA-miRPath): Mitogen-activated protein kinase (MAPK) signaling pathway, transforming growth factor (TGF)-β, Wnt (Wingless and Int-1), and signaling pathways related to various cancers such as pancreatic cancer and prostate cancer (Figure 2C). In addition, other pathways that are linked with infection and host cell associations, such as adherens junctions, and focal adhesion were also identified. MAPK signaling pathway containing *T. forsythia* significantly differentially upregulated miRNAs in mandibles compared with sham-infected controls at *p* < 0.05. *T. forsythia* significantly altered 45 gene expressions based on upregulated miRNA profiles (Figure 3A). Similarly, downregulated miRs involved with T-cell receptor signaling in the 8-week infection group were associated with MAPK signaling pathways, calcium signaling pathways, osteoclast differentiation signaling pathway, PI3K-Akt signaling pathways, NF-κB signaling pathways, cell adhesion molecules, and ubiquitin-mediated proteolysis genes (Figure 3B and Appendix A). Detailed reported functions of the upregulated miRNAs during 16 weeks of infection and downregulated miRNAs during 8 weeks of infection are shown in Table 3 and Table 4, respectively. Similarly, detailed reported functions of the downregulated miRNAs during 8 and 16 weeks of *T. forsythia* infection are shown in Appendix A. The number of target genes for each upregulated miRNAs in the 16-week infection group was analyzed using miRTarBase. We used mmu-miR-1902 as the example for an upregulated DE miRNA during 16 weeks of infection in identifying the target genes using miRTarBase (Appendix A). Each miRNA has different target genes along with a specific miRTarBase ID. *T. forsythia*-infection induced DE-upregulated mmu-miR-1902 has five different target genes with five different miRTarBase IDs as stated in Appendix A.

### 2.5. XGBoost Analysis of miRNA Copies

Using XGBoost, we analyzed which miRNAs were most important for predicting whether a mouse was infected with *T. forsythia*. For this analysis, we used the number of copies of each miRNA as our features and the infection status (whether the mouse was infected with *T. forsythia*) as our target. We used Scikit-learn’s RandomizedGridSearchCV method to determine which hyperparameter’s to use for the XGBoost model. The RandomizedGridSearchCV method was executed using between 2 and 11 cross-validation splits of the data, and the performance of each cross-validation split was evaluated using the LeaveOneOut method. The LeaveOneOut method was chosen be-cause of the small number of samples available in the data. After RandomizedGridSearchCV and LeaveOneOut determined the best hyper-parameters, we performed three analyses of the data: an analysis of the data for the mice infected at 8 weeks, one for the mice infected at 16 weeks, and a third analysis of the data for the combined group of mice infected at 8 and 16 weeks. Using the SHAP library, we then examined which miRNAs contributed most to the predictive model. For the 8-week dataset and the combined dataset (i.e., combining both the 8-week and 16-week datasets), miRNA mmu-miR-592-5p (MIMAT0003730) was determined to contribute the most to the model. For the 16-week dataset, mmu-miR-339-5p (MIMAT0000584) contributed the most. To mitigate unintended bias in the data, the feature order was randomly shuffled before performing the analysis. The results of the SHAP value analysis are summarized in Figure 4, and the results of the top five miRNA features for each dataset are summarized in Table 5.

### 2.6. Random Forest Machine Learning Model

Using Scikit-learn’s Random Forest methods, we analyzed the miRNAs that were most important for predicting whether a mouse was infected with *T. forsythia*. Similar to our XGBoost analysis (see Section 2.5), we used RandomizedGridSearchCV and LeaveOneOut to search for the best hyper-parameters, and then analyzed the results for predicting infected mice at 8 weeks, 16 weeks, and the combined 8-week and 16-week cohorts. Using the SHAP library, Random Forest determined that miR-592 (MIMAT0003730) was the most predictive for the 8-week and the combined cohorts (i.e., combining both the 8-week and 16-week datasets). This was the same as the XGBoost analysis. However, for the 16-week cohort, miR-423-5p (MIMAT0004825) was determined to be most predictive, which differed from the XGBoost analysis. To mitigate unintended bias in the data, the feature order was randomly shuffled before performing the analysis. The results of the SHAP value analysis are summarized in Figure 5, and the results of the top five miRNA features for each dataset are summarized in Table 6.

## 3. Discussion

Recent studies using partial human mouth microbes (PAHMM) in the ecological time-sequential polybacterial periodontal infection model (ETSPPI) showed sex-specific differential miRNA expression [20]. This PD mouse model utilized five different bacteria including *Streptococcus gordonii* (early colonizer), *F. nucleatum* (intermediate colonizer), *P. gingivalis*, *T. denticola* and *T. forsythia* (late colonizers) using a time sequential infection. Although we developed a highly efficient polybacterial-induced PD mouse model, studies on the functions of individual specific pathogens including well-characterized red complex bacteria such as *P. gingivalis*, *T. denticola*, *T. forsythia*, *F. nucleatum* in differential miRNA expression is slowly evolving. In this context, we reported several specific signature miRNAs during monoinfection with *P. gingivalis* [21] and *T. denticola* [22]. The present study aimed to investigate the potential role of tissue miRNAs as novel biomarkers of the *T. forsythia* infection to better understand the induction process of individual bacteria in periodontal disease. Periodontitis is a polymicrobial dysbiosis and the accumulation of subgingival biofilms rather than the sole influence of a single microbe. We analyzed *T. forsythia* colonization on the mouse gingival surface, horizontal ABR measurements, intravascular dissemination of *T. forsythia* to distal organs and global miRNA profiling in *T. forsythia* mice infected intraorally at 8 weeks and 16 weeks. All the mice infected with *T. forsythia* at both time points showed a 100% bacterial colonization on the gingival surface which was confirmed with 16S rRNA gene amplification. Significantly higher ABR was observed in *T. forsythia* infected mice at both time-points. This data was in accordance with our previously published data where significantly higher ABR was observed in ApoE^−/−^ mice infected with *T. forsythia* through intraoral infection for eight infection cycles over a period of 12 and 24 weeks [11].

This study profiled 577 mouse miRNAs in mandibles from *T. forsythia*-infected mice and sham infection using the NanoString nCounter system. A total of 115 miRNAs were DE in mandible tissue during 8- and 16-weeks *T. forsythia* infection compared with sham infection, and the majority (99) of DE miRNAs were downregulated. A large number of downregulated miRNAs indicate massive alterations in the biological processes involved in the pathogenesis of PD. miRNA profiling of mouse mandibles infected with *T. forsythia* showed unique miRNA gene signature in both 8-week and 16-week infections. Among the 16 upregulated miRNAs in 16-week *T. forsythia*-infected mice, 8 of the miRNAs were shown to be observed in human periodontitis patients indicating its robust miRNA genesis in induction of PD, miR-423-5p (upregulated in severe PD) [26], miR-210 (upregulated in PD) [29], miR-146a (overexpressed in saliva of patients with periodontitis) [31,32,33], miR-let-7a (upregulated in chronic periodontitis) [35], miR-24 (upregulated in the inflamed gingival biopsies, saliva of chronic periodontitis) [27], miR-218 (decreased expression is associated with protective effect in periodontitis) [42], miR-26b (downregulated in periodontal inflammation) [54], and miR-23a-3p [98]. In addition, eight DE miRNAs were associated with many systemic diseases, suggesting the miRNAs role in linking *T. forsythia* with periodontitis and systemic diseases—mmu-let-7c (human osteoarthritis and rheumatoid arthritis) [24], miR-423-5p (diabetes mellitus and AD) [28], miR-210 (atherosclerosis [30] and multiple cancers), miR-127 (upregulated in human atherosclerotic plaques) [36], miR-98 (cardiac hypertrophy) [37], miR-24 (coronary artery diseases [38] and oral squamous cell carcinoma) [40], miR-218 (atherosclerosis) [43], miR-101b (Alzheimer’s disease) [45]. All 16 of the identified DE miRNAs were reported to be observed for the first time in *T. forsythia*-induced periodontitis. The observation of these miRNAs in human periodontal gingival tissues, *T. forsythia*-induced periodontitis, and several systemic diseases highlights the importance of further in-depth analysis to understand these miRNAs as potential biomarkers for *T. forsythia*-induced periodontitis.

We have shown previously, in our combined in vitro and in vivo study, that THP-1 cells infected with live and heat-killed *T. forsythia* were the most potent inducers of IL-1β and TNF compared to polymicrobial infection [99]. As THP-1 monocytes are frequently used to examine innate immune ligand-induced miRNA expression, we observed significantly increased miR-132 expression with live *T. forsythia* in this study. The most widely investigated inflammatory miRNA in PD is miR-146a, which was upregulated in 16-week *T. forsythia*-infected mice as well as overexpressed in saliva of patients with PD [31,100]. It is obvious that the upregulated miRNAs in the current study may also have these functions as it is involved in NF-κB activation for transcription of inflammatory genes. Interestingly, the upregulated miRNA-101b was found to be an important mediator of tauopathy and dendritic abnormalities in AD progression [45]. These reports strongly suggest miRNA-101b, miRNA-let-7c, and miR-423-5p as a possible link between *T. forsythia*-induced gingival, synovial, and cerebral inflammation.

The KEGG pathway analysis revealed that most of the target genes of these miRNAs that were upregulated during 8-week and 16-week *T. forsythia* were linked with the MAPK signaling pathway, the TGF-β signaling pathway, the Wnt signaling pathway, and the pathways related to various cancers such as pancreatic andprostate cancer. Furthermore, 13 miRNAs were found to be closely associated with the MAPK signaling pathway that significantly altered 45 gene expressions based on upregulated miRNA profiles. Similarly, most of the pathways targeted by the miRNAs that were downregulated during 8 weeks of *T. forsythia* were linked with the T cell receptor signaling pathway that altered 31 genes. A recent clinical study identified functional circRNAs and the prediction of the circRNA-miRNA-mRNA regulatory network in periodontitis [101]. Our data show that monoinfection with *T. forsythia* can induce periodontitis in mice, although how PD is initiated is not completely understood. Several questions remain unanswered although the role of *T. forsythia* alone in the induction PD in an experimental system cannot be discounted. In addition, seven of the miRNAs in which *T. forsythia* induced DE are also observed in human gingival and saliva samples of patients with periodontitis. Furthermore, in-depth analysis on the identified DE miRNAs has the potential to reveal that they serve as tissue invasive bacterial biomarkers of pathogenesis using several knockout mouse models, providing more details on disease pathogenesis and its link with other systemic diseases.

The XGBoost analysis of which miRNAs (i.e., features) were most relevant for predicting whether a mouse was infected with *T. forsythia* revealed that miR-592-5p and miR-339-5p are contributory outcomes in 8- and 16-week analyses, respectively. However, a limitation of this study is that these two miRNAs were not unique to *T. forsythia* infection. This is one of the fundamental approaches to further investigate these miRNAs in the clinical setting. One study analyzed miRNA data with XGBoost and predicted miRNAs miR-152-3p, miR-221-3p, and miR-34a-5p in neurological outcomes of patients with subarachnoid hemorrhage [102].

The Random Forest analysis highlights miR-592 and miR-423-5p miRNAs as the most important features in 8- and 16-week analyses, respectively. miR-339-5p, miR-423-5p, and miR-1892 were the conserved features in the 16-week output of XGBoost analysis. The Random Forest algorithm identified 30 miRNA features in the pancreatic cancer [103] and in diabetes mellitus [104].

## 4. Materials and Methods

### 4.1. Induction of Periodontitis in C57BL/6J Mouse Using Intraoral Infection

*T. forsythia* ATCC 43037 (*Tf*) was grown in tryptic soy broth supplemented with N-acetyl muramic acid (5 mg/mL) and hemin (1 mg/mL) [20,99,105,106] for three days in an anaerobic growth chamber. Both male and female mice were divided into four groups (*n* = 10) (Group-I: *Tf*-infected for 8 weeks; Group-II: sham-infected for 8 weeks; Group III: *Tf*-infected for 16 weeks; Group-IV: sham-infected for 16 weeks) (Figure 1A). Kanamycin (500 mg/mL) was administered to all mice in sterile drinking water for three days to suppress the mouse oral bacteria followed by rinsing with a 0.12% chlorhexidine gluconate (Peridex: 3M ESPE Dental Products, St. Paul, MN, USA) [20,99,105,106]. After the antibiotic washout period, mice in Groups I and III were infected with *T. forsythia* (10^8^ cells each) suspended in equal volumes of reduced transport fluid (RTF) and carboxymethylcellulose (CMC) by intraoral infection. An equal volume of RTF and CMC was used as a vehicle control for mice in Groups II and IV as described previously [20,103,107,108]. Four infection cycles, as described, were performed for Group I and Group II mice. Eight infection cycles, as described, were performed for Group III and Group IV mice. One infection cycle consists of four days per week of *T. forsythia* intraoral infection/vehicle control suspension for every alternative week (Figure 1A). The mice were euthanized at their respective time points, and their mandibles, maxilla, and distal organs—brain, heart, liver, lungs, spleen and kidney were collected. Left maxilla and mandibles were collected in RNAlater for miRNA analysis, whereas right maxilla and mandibles were used for alveolar bone morphometry measurements. All the mouse procedures were performed according to the guidelines (University of Florida Institutional Animal Care and Use Committee protocol number 202200000223). This preclinical study complied with the ARRIVE guidelines (Animal Research: Reporting In Vivo Experiments).

### 4.2. Gingival Plaque Sample Analysis and Bacterial Dissemination to Distal Organs

Mouse gingival surface was swabbed in a sterile veterinary cotton swab and placed in 150 µL of sterile TE buffer after infection. *T. forsythia* genomic DNA was detected using 16S rRNA gene-specific primers as described previously [20,104]. *T. forsythia*-specific 16S rRNA gene-specific forward primer 5′-AAAACAGGGGTTCCGCATGG-3′ and reverse primer 5′-TTCACCGCGGACTTAACAGC-3′ were used to amplify the 16S rRNA gene in the gingival plaque samples. Genomic DNA from distal organs such as the brain, heart, liver, lungs, spleen and kidney were extracted using a standard protocol described in the Qiagen Dneasy Blood and Tissue kit (Qiagen, Germantown, MD, USA). *T. forsythia*-specific 16S rRNA gene amplification was performed to identify the *T. forsythia* genomic DNA as described previously [20,104].

### 4.3. Alveolar Bone Resorption (ABR) Measurements of Mandibles and Maxilla

The horizontal ABR was measured in right mandibles (lingual) and right maxilla (buccal and palatal) as described previously [20,21]. After euthanizing the mice, the jaws were placed in a beaker followed by autoclaving to remove the flesh surrounding the jaw bone. The mouse jaw bones were bleached with 3% hydrogen peroxide for 30 min and the samples were air dried. Two-dimensional imaging was performed using stereo dissecting microscope (Stereo Discovery V8, Carl Zeiss Microimaging, Inc., Thornwood, NY, USA). A line tool was used to measure the horizontal ABR between cementoenamel junction and alveolar bone crest (AxioVision LE 29A 4.6.3, Thornwood, NY, USA) [20,104]. Two examiners blinded to the study mice groups measured the ABR.

### 4.4. NanoString nCounter miRNA Panel and Data Analysis

The total RNA from the left mandibles (*n* = 10 mice per group; 5 males and 5 females) of *T. forsythia*-infected and sham-infected group was extracted using the miRVana Isolation kit (Ambion, Austin, TX, USA) as described previously [21,22]. RNA was quantified using the Take3 micro-volume plate in the Epoch Microplate Spectrophotometer (BioTek, USA, Winooski, VT, USA). As per the recommendation, RNA with the OD 260/230 ratio of >1.8 and the OD 260/280 ratio of >2 was taken for NanoString analysis. High-throughput nCounter^®^ miRNA Expression Panels (NanoString Technologies, Seattle, WA, USA) were used to examine the DE of miRNAs in 8- and 16-week *T. forsythia*-infected and sham-infected male and female mice as described in our recent publications [21,22]. The NanoString nCounter^®^ panel identifies a total of 577 mouse miRNAs in any RNA specimen, and by using molecular barcodes, NanoString can detect even a small number of miRNAs without the need for reverse transcription or amplification. Accordingly, there is no need to validate the NanoString data using stem-loop Real-Time PCR. Additionally, independent studies support the finding that there is no need to validate the NanoString data using stem-loop Real-Time PCR [109,110]. Mouse miRNA expression profiling was performed using the NanoString nCounter^®^ Mouse miRNA Assay kit v1.5. This assay is a highly sensitive multiplexed method that detects miRNAs using molecular barcodes called nCounter reporter probes. Specimen preparation involving annealing, ligation, and purification were performed based on the experimental procedure described in the miRNA assay panel kit and described in our recent reports [21]. In order to obtain the unbiased data, stringent quality check was performed. Data were analyzed and normalized using nSolver™ 4.0. Software Analysis (NanoString Technologies, Seattle, WA, USA). All the samples passed the quality check, and in order to reduce the background signal, background count threshold was set up at 52. NanoString data were used to obtain the miRNA expression in each sample as a fold-change value, and a fold change of ± 1 was considered significant. Data normalization was performed based on the top 100 miRNAs expressed in each sample.

### 4.5. Bioinformatic Analysis

All the normalized data were analyzed further using ROSALIND (https://rosalind.bio/, accessed on 1 May 2023), with a HyperScale architecture developed by ROSALIND, Inc. (San Diego, CA, USA) [111]. Fold changes between the groups were measured based on the ratio of difference in the means of log-transformed normalized data to the square root of the sum of the variances of each sample in the respective groups. For Kyoto Encyclopedia of Genes and Genomes (KEGG) pathway analysis, we analyzed the DE miRNAs in 8-week and 16-week *T. forsythia*-infected mouse mandibles in the DIANA-miRPath v.3.0 database [112]. This software links miRNAs to target genes from Tarbase, v7.016. All the DE miRNAs were entered using the MIMAT accession number in the DIANA-miRPath database with the threshold values of *p* < 0.05 and false discovery rate (FDR) correction applied to obtain unbiased empirical distribution using the Benjamin and Hochberg method. Additionally, we analyzed the DE miRNAs in 8- and 16-week *T. forsythia*-infected mice mandibles in miRTarBase which is the experimentally validated microRNA-target interactions database (miRTarBase update 2022: an informative resource for experimentally validated miR-NA-target interactions) [113]. A Venn diagram for higher expression and downregulated miRNAs in the 8-week and 16-week infection groups was drawn using Venny 2.1 [21].

### 4.6. Machine Learning Model Analysis

Machine learning analysis was performed on a 2021 MacBook Pro with 64 gigabytes of RAM and an Apple M1 chip. We used Python version 3.11.4, XGBoost version 1.7.6, and SHAP version 0.42.1 to obtain feature importance results [114,115]. NanoString analysis miRNA copy data from the 8- and 16-week *T. forsythia* infection groups were collected and restructured into a format suitable for analysis by XGBoost [116,117,118,119]. The combined analysis of the 8- and 16-week *T. forsythia*-infected mice was performed by simply merging the two miRNA copies datasets together. To address implicit biases in the data, the order of the rows and columns was randomly shuffled prior to analysis. The full code used to perform the analysis is available on Github at https://github.com/uflcod/miRNA-periodontal-disease/blob/main/notebooks/xgboost_miRNA.ipynb (accessed on 11 August 2023).

### 4.7. Random Forest Machine Learning Model

The algorithm of Random Forest classification was performed according to the multiple decision trees. Feature importance was determined by the mean decrease Gini Index calculated by Random Forest. The miRNAs output of NanoString analysis in 8 and 16 weeks of *T. forsythia*-infected mice and sham-infected 8- and 16-week mice used to address the issue of miRNAs as novel markers in the periodontitis. The Random Forest algorithm has the leverage in predicting miRNAs as markers in disease conditions [120,121].

### 4.8. Statistical Analysis

All the data in the graphs were presented as mean ± SEM. Ordinary two-way ANOVA with Turkey’s multiple comparison test with a single pooled variance was performed for ABR measurements to identify the statistical significance using Prism 9.4.1 (GraphPad Software, San Diego, CA, USA) [20,21]. A *p*-value of <0.05 was considered statistically significant. Identification of *T. forsythia*-induced differential gene expression was performed based on two-tailed *t*-tests on the log-transformed normalized data that assumed unequal variance. The distribution of the t-statistics was calculated using the Welch–Satterthwaite equation for the degrees of freedom to estimate the 95% confidence intervals for the identified DE of miRNA between infection and sham infection mice. The Volcano plot was drawn using the GraphPad Software (Prism 9.4.1) [20,21].

## 5. Conclusions

This is the first monobacterial preclinical in vivo study that delineated the miRNA kinetics of *T. forsythia* intraoral infection. Unique miRNAs were identified in both 8-week (downregulation: miR-375, miR-200b, c, miR-141, miR-423-3p) and 16-week *T. forsythia*-infected (higher expression: miR-1902, miR-423-5p, miR-210, miR-146a) mice. It is interesting to note that many miRs upregulations (miR-423-5p, miR-210, miR-146a) and downregulations (miR-200b, c, miR-141, miR-423-3p) to the *T. forsythia* infection were shown as potential biomarkers in patients with periodontitis. In addition, several upregulated miRNAs (e.g., miR101b, miR-218, miR-127, miR-24) were also associated with many systemic diseases such as atherosclerosis, Alzheimer’s disease, rheumatoid arthritis, osteoarthritis, diabetes, obesity, and several cancers. Since the exact molecular effects of these miRs are not presently known, their biological function and significance should be explored in follow-up studies. Five miRs (1902, 375, 200c, 200b, 141) and their DE with a log2-fold-change in the range of 1.72 to 2.38 in *T. forsythia* infection may be considered as novel pathogenic miRNA biomarkers. Accurate identification of biomarker miRNAs is important to further the understanding of various biological mechanisms. This study’s utilization of the XGBoost machine learning library identified miR-339-5p as most predictive for mice infected at 16 weeks, and miR-592-5p as most predictive for mice infected at 8 weeks as well as when the 16-week and 8-week cohorts were analyzed together. Similarly, the Random Forest ML learning library identified miR-2133 and miR-339-5p as the most important features in 8 and 16 weeks, respectively. To the best of our knowledge, this is the first study to investigate the expression of mouse gingival miRNAs using a wide panel of 577 miRNAs during *T. forsythia* infection-induced periodontitis. Future studies are needed to understand the molecular mechanism and miRNAs biomarkers involved in periodontal disease.

## Figures and Tables

**Figure 1 ijms-24-16393-f001:**
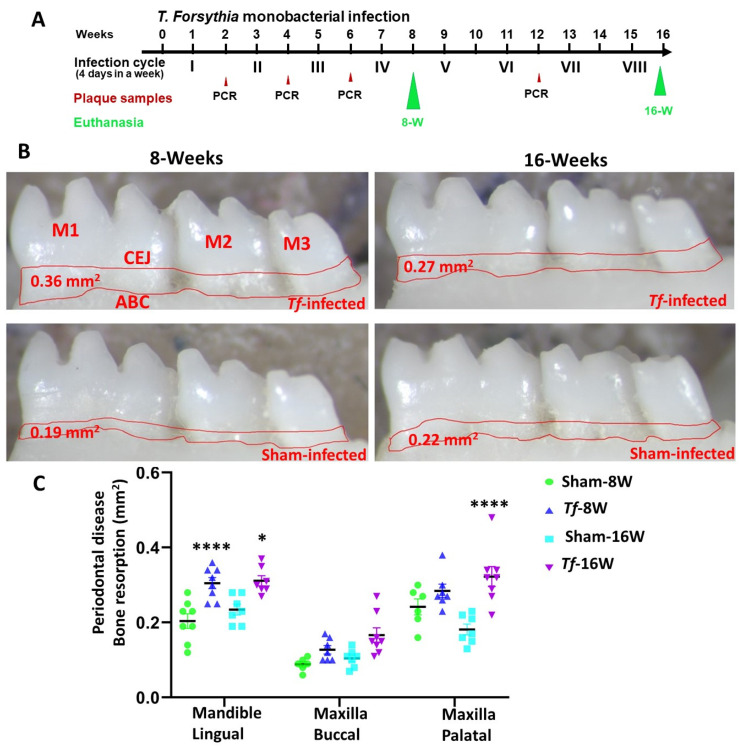
Intraoral infection of *T. forsythia* significantly induced alveolar bone resorption. (**A**) Schematic diagram of the experimental design depicting the monobacterial *T. forsythia* infection (4 days per week on every alternate week), plaque sampling for PCR and euthanasia. (**B**) Representative images showing horizontal ABR (mandible lingual view) of *T. forsythia*-infected and sham-infected mice with the area of alveolar bone resorption outlined from the alveolar bone crest (ABC) to the cementoenamel junction (CEJ). (**C**) Morphometric analysis of the mandible and maxillary ABR in mice. A significant increase in ABR was observed in *T. forsythia* -infected mice compared to sham-infected mice in both 8-week and 16-week-infected mice (****, *p <* 0.0001; *, *p* < 0.1, ordinary two-way ANOVA). Data points and error bars are mean ± SEM (*n* = 10).

**Figure 2 ijms-24-16393-f002:**
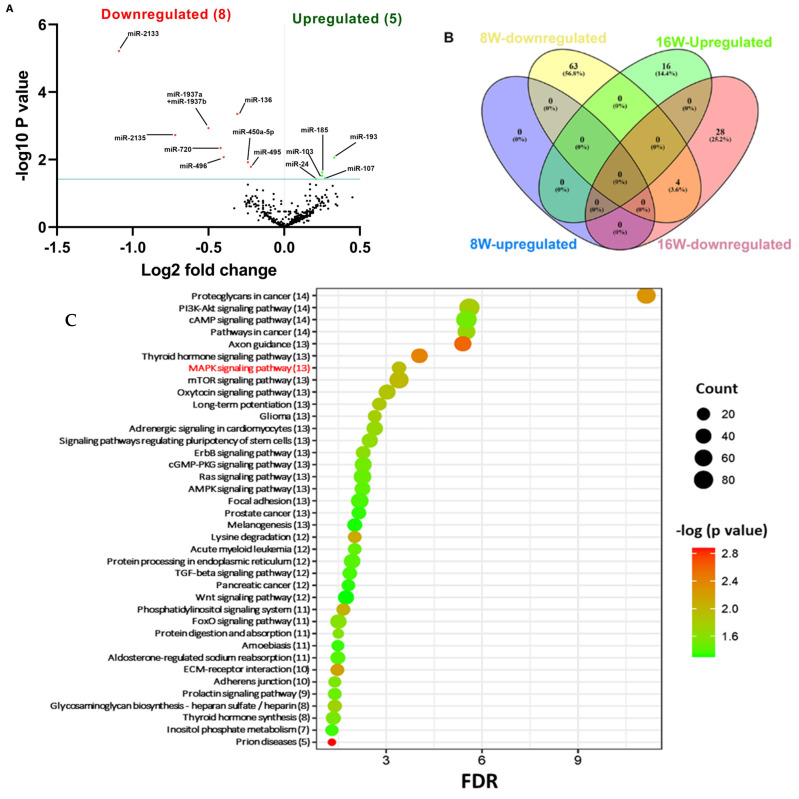
Differentially expressed miRNAs in *T. forsythia*-infected mandibles (8 and 16 weeks). (**A**) The volcano plot depicts the upregulated (green) and downregulated (red) miRNAs that showed a fold difference of ±1.1 with a *p*-value of <0.05 during *T. forsythia* infection. Fold changes (log2 fold change) are displayed on the *x* axis, the *y* axis shows the *t*-statistic derived as the negative decimal logarithm of the *p*-value [−log10(*p*)]. Thresholds are highlighted with dotted lines representing the promising miRNAs in the upper left and upper right. The black dots represent the miRNAs that do not pass the filter parameters. Five significantly upregulated miRNAs and nine downregulated miRNAs were identified in 16-week *T. forsythia*-infected mice compared to the 8-week *T. forsythia*-infected mice (*n* = 10). (**B**) Venn diagram analysis illustrates the distribution of DE miRNAs in the 8-week and 16-week infections with *T. forsythia*. (**C**) Predicted functional pathway analysis of DE miRNAs from *T. forsythia*-infected mandibles. Bubble plot of KEGG analysis on predicted target genes of DE-upregulated miRNAs in *T. forsythia*-infected mice (16-week duration) comparing sham-infected mice. The KEGG pathways are displayed on the *y* axis, and the *x* axis represents the false discovery rate (FDR) which means the probability of false positives in all tests. The size and color of dots represent the number of predicted genes and the corresponding *p*-values, respectively. Thirteen DE miRs are shown to be involved in the MAPK signaling pathway.

**Figure 3 ijms-24-16393-f003:**
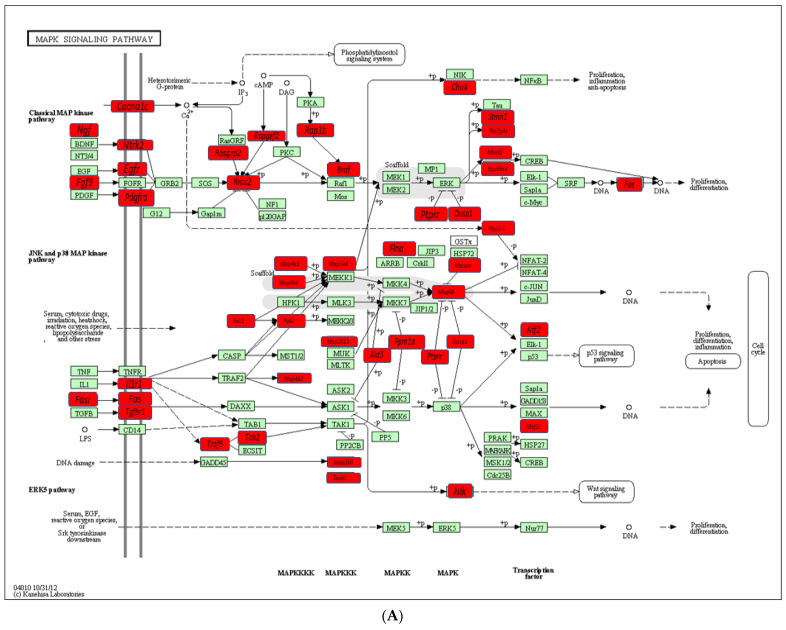
(**A**) MAPK signaling pathway containing *T. forsythia* upregulated miRNAs in mandibles compared with sham-infected controls at *p* ≤ 0.05, adapted from the pathway express and using KEGG. Red boxes indicate 45 upregulated miRNA profiles from the NanoString analysis. Green boxes indicate no change in gene expression. An arrow indicates a molecular interaction resulting in classical MAP kinase pathway, phosphatidylinositol signaling system, JNK and p 38 MAP kinase pathway, p53 signaling pathway, ERK5 pathway, Wnt signaling pathway. A line without an arrowhead indicates a molecular interaction resulting in inhibition. Solid and dash-dotted lines denote direct and indirect relationships, respectively. +P indicates phosphorylation, and −P indicates dephosphorylation. The MAPK cascade is a highly conserved module that is involved in various cellular functions, including cell proliferation, differentiation and migration. (**B**) T cell receptor signaling pathway containing *T. forsythia* downregulated miRNAs in mandibles compared with sham-infected controls at *p* ≤ 0.05, adapted from the pathway express and using KEGG. Red boxes indicate 31 significantly downregulated miRNA profiles from the NanoString analysis. Green boxes indicate no change in gene expression. An arrow indicates a molecular interaction resulting in MAPK signaling pathway, calcium signaling pathway, PI3K-Akt signaling pathway, NF-κB signaling pathway, and ubiquitin-mediated proteolysis. A line without an arrowhead indicates a molecular interaction resulting in inhibition.

**Figure 4 ijms-24-16393-f004:**
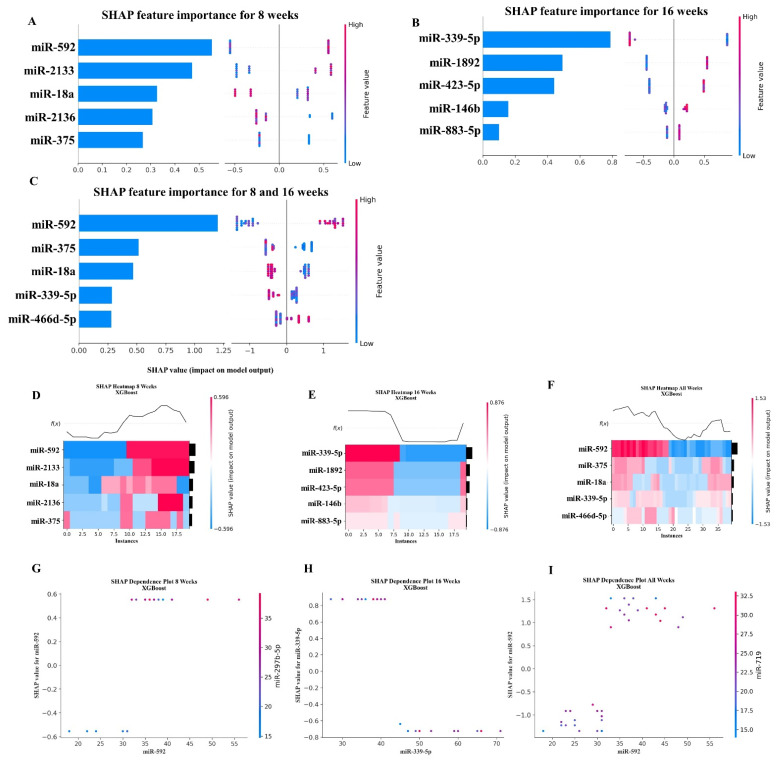
A summary of the most important features in the XGBoost model using SHAP values. In figures (**A**–**F**), feature importance is ranked from top (the most important) to bottom (the lease important). The *x* axis in (**A**–**C**) shows the impact that a feature has on the model. The bar charts show the overall impact of a feature whereas the swarm plot shows both the positive and negative impacts. In the swarm plots, each dot represents an instance of an miRNA variable and the color bar shows the value (high to low) of the variable. In (**A**), the results are shown for the mice tested at 8 weeks. In (**B**), the results from the mice tested at 16 weeks are shown, and (**C**) shows the results of analyzing both the 8- and 16-week cohorts together. In (**D**–**F**), the *x* axis represents each mouse (instance) in the cohort, and the *y* axis is the feature ranking. The color of the cell shows the amount of impact (i.e., the SHAP value) that particular feature inflicted on that feature. The topmost section (i.e., *f*(*x*)) of the heatmap shows the predicted infection status for each instance. In all three cohorts, there is strong correlation between high values for the topmost feature and the model predicting that the mouse was infected. In (**G**–**I**), the relation between the topmost feature and the feature it most depends on is shown. Each dot represents a mouse and the *x* axis shows the value of the miRNA variable. The left *y* axis shows the impact (i.e., the SHAP value) the *x* axis variable has on the mode. The right *y* axis shows the value of the variable that the *x* axis variable interacts with.

**Figure 5 ijms-24-16393-f005:**
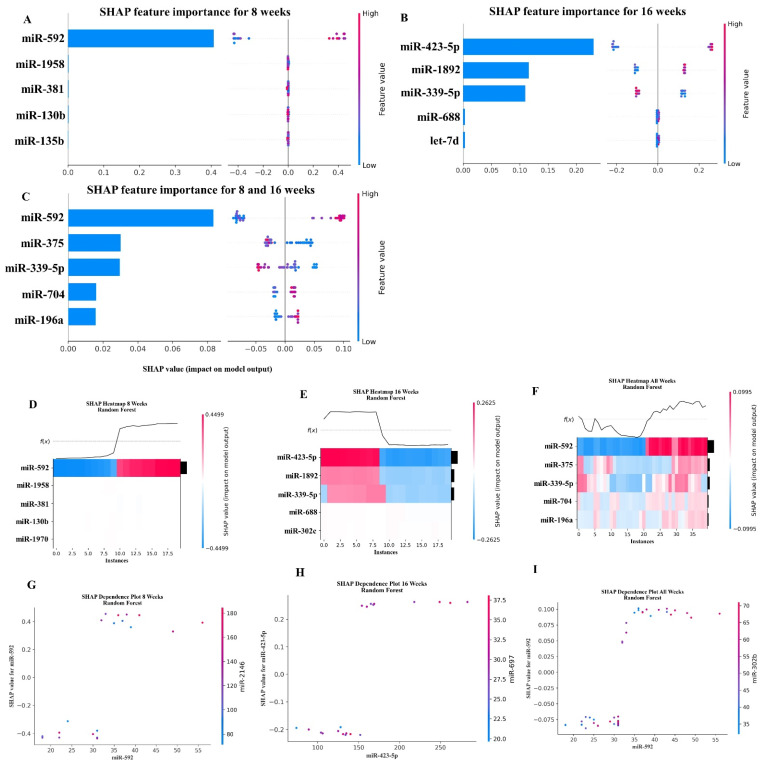
A summary of the most important features in the Random Forest model using SHAP values. In (**A**–**F**), feature importance is ranked from top (the most important) to bottom (the lease important). The *x* axis in (**A**–**C**) shows the impact that a feature has on the model. The bar charts show the overall impact of a feature whereas the swarm plot shows both the positive and negative impacts. In the swarm plots, each dot represents an instance of an miRNA variable and the color bar shows the value (high to low) of the variable. In (**A**), the results for the mice tested at 8 weeks are shown. In (**B**), the results from the mice tested at 16 weeks are shown, and in (**C**), the results of analyzing both the 8- and 16-week cohorts together are shown. In (**D**–**F**), the *x* axis represents each mouse (instance) in the cohort, and the *y* axis is the feature ranking. The color of the cell shows the amount of impact (i.e., the SHAP value) that particular feature inflicted on that feature. The topmost section (i.e., *f*(*x*)) of the heatmap shows what is the predicted infection status for each instance. In all three cohorts, there is strong correlation between high values for the topmost feature and the model predicting that the mouse was infected. In (**G**–**I**), the relation between the topmost feature and the feature it most depends on is shown. Each dot represents a mouse and the *x* axis shows the value of the miRNA variable. The left *y* axis shows the im-pact (i.e., SHAP value) the x axis variable has on the mode. The right *y* axis shows the value of the variable that the *x* axis variable interacts with.

**Table 1 ijms-24-16393-t001:** Gingival plaque samples tested positive for *T. forsythia* gDNA using PCR.

Group/Bacteria/Weeks	Positive Gingival Plaque Samples (*n* = 10)			
	8-Week-Time Point		16-Week-Time Point	
	2 Weeks	4 Weeks	6 Weeks	12 Weeks
Group I/*T. forsythia* ATCC 43037 (8 weeks)	5/10	8/10	---	---
Group II/sham infection (8 weeks)	0/10	NC	---	---
Group III/*T. forsythia* ATCC 43037 (16 weeks)	1/10	7/10	NC	10/10
Group IV/sham infection (16 weeks)	0/10	NC	NC	0/10

Total numbers of gingival plaque samples that were collected after *T. forsythia* infections (2, 4, 6 and 12 weeks) and were positive as determined by PCR analysis. NC—not collected to allow bacterial biofilm adherence to gingival surface, invasion of epithelial cells, and multiplication. The first value corresponds to the number of mice that tested positive for *T. forsythia* bacterial genomic DNA and the second value corresponds to the total number of mice in the group.

**Table 2 ijms-24-16393-t002:** Differentially expressed miRNAs during 8- and 16-week *T. forsythia* infections in mice.

Weeks/Infection/Sex	Upregulated miRNAs (*p* < 0.05)	Downregulated miRNAs (*p* < 0.05)
8 Weeks/*T. forsythia*-infected vs. 8 Weeks/sham-infected (*n* = 10)	0	67 (miR-375, miR-200c, miR-200b, miR-141, miR-34b-5p)
8 Weeks/*T. forsythia*-infected Female vs. Male (*n* = 5)	15	21
16 Weeks/*T. forsythia*-infected vs. 16 Weeks/sham-infected (*n* = 10)	16 (miR-1902, miR-let-7c, miR-146a, miR-423-5p)	32 (miR-2135, miR-720, miR-376c, miR-488, miR-322)
16 Weeks/*T. forsythia*-infected Female vs. Male (*n* = 5)	1	10
8 Weeks/*T. forsythia*-infected vs. 16 Weeks/*T. forsythia*-infected	5	8

The number of DE miRNAs is shown for *T. forsythia* infected mice after 8- and 16-week infections. The significantly expressed top miRNAs between 8-week and 16-week *T. forsythia*-infected mice are shown in parenthesis. All of the DE miRNAs expressed in the *T. forsythia*-infected group were unique and specific to the 8- and 16-week infections.

**Table 3 ijms-24-16393-t003:** *T. forsythia* infection-induced upregulated miRNAs, reported functions and target genes.

Upregulated miRNAs in a 16-Week *T. forsythia* Infection				
miRNAs	Fold Change	*p*-Value	Reported Functions	Number of Target Genes
miR-1902	1.67	0.00313	Overexpressed in mouse serum and whole blood after intraperitoneal injection of lipoteichoic acid [23].	--
mmu-let-7c	1.39	0.0045	Interfere with critical inflammatory cytokine production (IL-1β, IL-6, and TNF) in human osteoarthritis and rheumatoid arthritis [24]. Playing a role in cardiomyogenesis promotion activity [25].	1782 (e.g., *Fyco1*, *ADH4*, *Socs1*, *Steap4*, *S100pbp*)
miR-423-5p	1.32	4.7 × 10^−7^	Upregulated expression in severe periodontal disease [26]. Higher expression in obese periodontitis subjects [27]. Identified as new candidate biomarker in the cross-talk between diabetes mellitus and Alzheimer’s disease [28].	8 (e.g., *Smtnl2*, *Rnf114*, *Prune*, *Mcl1*, *Pura*)
miR-210	1.31	0.02324	Upregulated in periodontal disease and obesity-suffering subjects [29]. Elevated in the muscle samples of peripheral artery diseases and atherosclerosis obliterans [30]. Frequently elevated in multiple cancers such as HCC, prostate cancer, colorectal cancer and gastric cancer.	427 (e.g., *Fyco1*, *Arl8b*, *Tpm3*, *Gcnt4*, *P2rx7*)
miR-146a	1.3	0.01937	Overexpressed in saliva of patients with periodontitis and its expression is increased with the deterioration of periodontitis in the patients [31,32]. Elevated levels are associated with reduction in proinflammatory cytokines in aggressive periodontitis [33].	360 (e.g., *Srrm2*, *Cpt1a*, *Calu*, *BC030336*, *Maff*)
miR-99b	1.28	0.00041	Upregulated in *M. tuberculosis* infected murine dendritic cells [34].	4 (e.g., *Hnrnpu*, *Tcf7l2*, *Trim71*, *Ctnnd1*)
mmu-let-7a	1.26	0.00595	Significantly upregulated in chronic periodontitis patients and interact with NF-κB pathway genes [35].	1038 (e.g., *Fyco1*, *S100pbp*, *Foxo1*, *Snx5*, *Igf1*, *Cxxc5*, *Rxra*)
miR-127	1.25	0.04454	Upregulated in human atherosclerotic plaques with similar expression of RTL1 [36].	4 (e.g., *Prx*, *Kpna2*, *Atf4*, *Hsp90ab1*)
miR-98	1.24	0.04528	Cardiac hypertrophy can be inhibited by upregulation of thioredoxin 1 which elevates the levels of miR-98 [37].	586 (e.g., *Srrm2*, *Vps26b*, *Vps54*, *Cep97*, *Snx25*, *Prdm1*)
miR-24	1.23	0.00075	Upregulated in inflamed gingival biopsies [27]. Decreased levels reported in coronary artery diseases [38]. Potential therapeutic target candidate for Human papillomavirus-mediated carcinoma [39]. Elevated levels in oral squamous cell carcinoma [40].	359 (e.g., *Fyco1*, *Mxd1*, *Flcn*, *Cblb*, *Bcl2l1*, *Cltc*)
miR-876-3p	1.23	0.01747	Reduces tumor cell growth and suppressor at its elevated levels [41].	---
miR-218	1.19	0.01142	Decreased expression is associated with protective effect in periodontitis [42]. Reduced cardiomyocyte hypertrophy. Reduced expression is a clinical marker for atherosclerosis [43]. Inhibition of miR-218 results in the attenuation of synovial inflammation and cartilage injury in the knee osteoarthritis rat model [44].	1139 (e.g., *Epg5*, *Dmxl1*, *Cxcr4*, *Socs3*, *Rassf3*, *Acly*, *Fyco1*)
miR-101b	1.17	0.04246	Major mediator of tauopathy and dendritic abnormalities in Alzheimer’s disease progression [45].	230 (e.g., *Cd55*, *Brd8*, *Dhx9*, *Tcp1*, *Cltc*, *Fos*, *Tes*)
miR-23b	1.14	0.00697	miR-23b mediates TNF-inhibited osteogenic differentiation of human periodontal ligament stem cells [46].	784 (e.g., *Sla*, *Fbxo5*, *Nus1*, *Pvr*, *Chd1*, *Igf1*, *Tle4*)
mmu-let-7e	1.14	0.01353	Inhibits the expression of collagen and post-transcriptional repression in HL1 cardiomyocytes [47].	1011 (e.g., *Tyk2*, *Lrp10*, *Eng*, *Smg7*, *Prc1*, *Psd3*, *Irs2*)
miR-26b	1.14	0.02741	Downregulated in periodontal inflammation [48].	1751 (e.g., *Sema3d*, *Inpp5d*, *Urb2*, *Coro7*, *Actr6*, *Arf6*)

Details of the target genes are given for the top five significantly upregulated DE miRNAs during the 16-week *T. forsythia* infection. miRNAs (miR-423-5p, miR-210, miR-146a, miR-let-7a, miR-24, miR-218, miR-24b) colored in bold red are associated with chronic periodontitis. miRNAs (e.g., miR101b, miR-218, miR-127, miR-24) that are associated with many systemic diseases (atherosclerosis, Alzheimer’s disease, rheumatoid arthritis, osteoarthritis, diabetes, obesity, and several cancers) are colored in bold blue.

**Table 4 ijms-24-16393-t004:** Downregulated miRNAs, reported functions, and target genes.

Dysregulated miRNAs in an 8-Week *T. forsythia* Infection				
miRs	Fold Change	*p*-Value	Reported Function	Number of Target Genes
miR-375	−2.38	0.0154379	Down-regulated in oral squamous cell carcinoma [49] and salivary adenoid cystic carcinoma [50].	24 (e.g., *Med13*, *C1qbp*, *Mtpn*, *Wdr26*, *Sept2*)
miR-200c	−2.36	0.01351086	Reduced levels observed in mice infected with LPS of *P. gingivalis* [51].	15 (e.g., *Atp5b*, *Pls3*, *Mycn*, *Zeb1*, *Thap4*)
miR-200b	−1.86	0.00822482	Variations in the levels of miR-200b observed in gingival tissue of obese periodontitis subjects [52].	100 (e.g., *Ythdf3*, *N4bp2*, *Apobec3*, *G6pc*, *Sc5d*)
miR-34b-5p	−1.8	0.02779496	Enhances the resistance to bleomycin by regulating its target gene TIMP3 during the pathogenesis of lung fibrosis [53].	14 (e.g., *Mapk1*, *Mycn*, *Pou5f1*, *Sox2*, *Tfcp2l1*)
miR-141	−1.72	0.01359321	Decreased levels reported in inflamed gingival tissues of periodontitis patients [54].	143 (e.g., *Dlc1*, *Apob*, *Acox2*, *Helz2*, *Klf5*)
miR-140	−1.48	0.03988881	Downregulated in cancer stem cells of DCIS tumors [55].	
miR-129-5p	−1.45	0.01918338	Downregulation fosters epithelial to mesenchymal transition in breast cancer [56].	
miR-205	−1.41	0.03628375	Decreased levels reported in inflamed gingival tissues of periodontitis patients [54].	
miR-423-3p	−1.4	0.00303573	Upregulated expression in severe periodontal disease [26]. Reduced expression reported in obese periodontitis subjects [27].	

Details of the target genes are given for the top five downregulated significantly expressed miRNAs in 8-week *T. forsythia*-infected mouse mandibles. miRNAs (miR-200b, miR-141, miR-205, miR-423-3p) colored in red are associated with periodontal disease.

**Table 5 ijms-24-16393-t005:** XGBoost analysis of predicted miRNAs and reported functions.

miRNA	Accession #	Target Function
**8-Week Analysis**		
miR-592	MIMAT0003730	Increased expression of miR-592 during expansion of rat dental pulp stem cells and their implication in osteogenic differentiation [57].
miR-2133	MIMAT0011209	Upregulated in mouse optic nerves exposed to oxygen glucose deprivation-treated MS-275 anticancer drug [58]. Downregulated in VCG vaccine immunized and *Chlamydia*-infected mice [59]. Reported in cardiac myocytes (Day 10) compared to normal P19 cells (Day 0) [60].
miR-18a	MIMAT0000528	Upregulated in obesity and associated periodontal disease condition [29]. Upregulated in periodontal ligament stem cells [61]. Associated with aortopathy patients [62]. Over-expression reduced fibrosis, hypertrophy, and apoptosis of cardiomyocytes in heart failure [63]. Elevated in the heart tissue of old age mice [64,65]. Aberrantly overexpressed in female CHD patient’s peripheral blood [66].
miR-2136	MIMAT0011212	Upregulated in mouse brains with Alzheimer’s disease (PMID: 29057267).
miR-375	MIMAT0000739	Down-regulated in oral squamous cell carcinoma [49] and salivary adenoid cystic carcinoma [50].
**16-Week Analysis**		
miR-339-5p	MIMAT0000584	Downregulated in osteogenic differentiation conditions of human bone marrow mesenchymal stem cells and has a role in the NEAT-1-miR-339-5p-SPI1 feedback loop [67]. Potential biomarker for multiple system atrophy patients and Parkinson’s disease [68]. Upregulated during neonatal rat cardiomyocytes hypertrophy triggered by isoproterenol [69]. Upregulated in exercise non-responders and involved in angiogenesis, skeletal muscle function and inflammation [70]. Upregulated in animal models of left ventricular ischemia [71].
miR-1892	MIMAT0007871	Reported in cardiac myocytes (Day 10) compared to normal P19 cells (Day 0) [60].
miR-423-5p	MIMAT0004825	Overexpression of miR-423-5p induced breast cancer cell invasion through the NF-κB signaling pathway [72].
miR-146b	MIMAT0003475	Elevated with the progression of periodontal disease [73]. Potential biomarker in periodontal disease and diabetes [74].
miR-883-5p	MIMAT0004848	Associated with neurotrophic pain and aging [75]. Bind to Cyp3a mRNAs and regulate Cyp gene expressions [76]. Increased in high-iodine exposed Wistar rats [77].
**Combined 8 and 16 Weeks**		
miR-592	MIMAT0003730	Shown in the 8-week analysis
miR-375	MIMAT0000739	Shown in the 8-week analysis
miR-18a	MIMAT0000528	Shown in the 8-week analysis
miR-339-5p	MIMAT0000584	Shown in the 16-week analysis
miR-466d-5p	MIMAT0004930	Upregulated in rat alveolar epithelial cells [78]

**Table 6 ijms-24-16393-t006:** Random Forest analysis-predicted miRNAs and reported functions.

miRNA	Accession ^#^	Target Function
**8-week analysis**		
miR-592	MIMAT0003730	Increased expression of miR-592 during expansion of rat dental pulp stem cells and their implication in osteogenic differentiation [57].
miR-1958	MIMAT0009431	Promoting *Mycobacterium tuberculosis* survival in RAW 264.7 cells [79].
miR-381	MIMAT0000746	Upregulated in saliva samples of periodontitis patients [80]. Associated with chronic periodontitis [81]. Protective role in coronary heart disease patients [82]. miR-381 as novel vehicles for promoting the osteogenic differentiation of BMSCs via Mg^2+^ ions [83].
miR-130b	MIMAT0000387	Remarkably improved cardiac function and ameliorated morphological damage to heart tissue in LPS-induced mice [84]. Upregulation of cfa-miR-130b was observed in dogs with myxomatous mitral valve degeneration [85]. Contributing in the prediction of T2DM patients with CVD [86].
miR-135b	MIMAT0000612	Significantly associated with myocardium adipose and fibrosis in the primary cardiomyopathy patients [87]. Downregulated in the bone microvascular endothelial cells isolated from non-traumatic ONFH [88].
**16-week analysis**		
miR-423-5p	MIMAT0004825	Overexpression of miR-423-5p induced breast cancer cell invasion through NF-κB signaling pathway [72].
miR-1892	MIMAT0007871	Reported in cardiac myocytes (Day 10) compared to normal P19 cells (Day 0) [60].
miR-339-5p	MIMAT0000584	Suppresses the invasion and migration of pancreatic cancer cells [89]. Promoting osteogenic differentiation [90]. Downregulated in metastatic gastric cancer patients [91], prostate cancer [92].
miR-688	MIMAT0003467	Protective factor in acute kidney injury [93]. Expressed in the central nervous system of mice models [94].
let-7d	MIMAT0000383	Attenuates epithelial–mesenchymal transition in silica-induced pulmonary fibrosis [95].
**8- and 16-week combined analysis**		
miR-592	MIMAT0003730	Increased expression of miR-592 during expansion of rat dental pulp stem cells and their implication in osteogenic differentiation [57].
miR-375	MIMAT0000739	Down-regulated in oral squamous cell carcinoma [49] and salivary adenoid cystic carcinoma [50].
miR-339-5p	MIMAT0000584	Suppresses the invasion and migration of pancreatic cancer cells [89]. Promoting osteogenic differentiation [90]. Downregulated in metastatic gastric cancer patients [91], prostate cancer [92].
miR-704	MIMAT0003494	Discovered in the late period of bleomycin-induced pulmonary fibrosis [96].
miR-196a	MIMAT0000518	Downregulated in the gingival tissue of obese periodontitis subjects [52]. Plays a role in immunity development [97]. The expression level of miR-196a in gingival sulcus was significantly higher in the periodontitis.

^#^: Accession Number.

## Data Availability

The data that support the findings of this study are openly available in NCBI https://www.ncbi.nlm.nih.gov/geo/query/acc.cgi?acc=GSE239421 (accessed on 27 July 2023).

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
