# Peer review of "Unique miRomics Expression Profiles in Tannerella forsythia-Infected Mandibles during Periodontitis Using Machine Learning"

_ijms, 2023, doi:10.3390/ijms242216393_

Round 1

Reviewer 1 Report

Comments and Suggestions for Authors

Comments on the IJMS MS ID 2641428 entitled ‘Unique miRomics expression profiles in Tannerella forsythia-in- 2 fected mandibles during periodontitis using machine learning." written by Chairmandurai Aravindraja, Syam Jeepipalli, William Duncan, Krishna Mukesh Vekariya, Edward K. L. Chan and Lakshmyya Kesavalu

I have read the submitted research report carefully, and here I am offering my assessment and comments.

The authors studied T. forsythia, a subgingival periodontal bacterium constituting the subgingival pathogenic polymicrobial milieu during periodontitis (PD). They stated that miRNAs are pivotal in maintaining periodontal tissue homeostasis at the transcriptional, post-transcriptional, and epigenetic levels. Their study aims to characterize the global microRNAs (miRNA, miR) expression kinetics in 8- and16-weeks T. forsythia-infected C57BL/6J mouse mandibles and to identify the miRNA bacterial biomarkers of the disease process at specific time-points. The authors examined the differential expression (DE) of miRNAs in mouse mandibles (n=10) using high throughput NanoString nCounter® miRNA expression panels, which provided significant advantages over specific candidate miRNA or pathway analyses. All the T. forsythia-infected mice at two distinct time points showed bacterial colonization (100%) in the gingival surface, along with a significant increase in alveolar bone resorption (ABR) (p < 0.0001). They performed NanoString analysis of specific miRNA signatures, miRNA target pathways, and gene network analysis. They found that a total of 115 miRNAs were DE in mandible tissue during 8- and 16-week T. forsythia infection as compared with sham infection, and the majority (99) of DE miRNAs were downregulated. nCounter miRNA expression-kinetics identified 67 dysregulated miRNAs (e.g. miR-375, miR-200c, miR-200b, miR-34b-5p, miR-141) during 8-week infection, whereas 16 upregulated miRNAs (e.g. miR-1902, miR-let-7c, miR-146a) and dysregulated miRNAs (e.g. miR2135, miR-720, miR-376c) were identified during 16-weeks infection. Two miRNAs miR-375, miR- 200c were highly dysregulated with >2-fold change during 8-week infection. Six miRNAs in the 8- week infection (miR-200b, miR-141, miR-205, miR-423-3p, miR-141-3p, miR-34a-5p) and two miRNAs in the 16-week infection (miR-27a-3p, miR-15a-5p) that were dysregulated have also been reported in gingival tissue and saliva of periodontitis patients. This preclinical in vivo study identified T. forsythia-specific miRNAs (miR-let-7c, miR-210, miR-146a, miR-423-5p, miR-24, miR-218, miR-26b, miR-23a-3p) and these miRs have also been reported in the gingival tissues and saliva of periodontitis patients. In addition to DE analysis, the authors utilized the machine learning (ML) library XGBoost (eXtreme Gradient boost) to assess the impact that the number of miRNA copies has on predicting if a mouse was infected. Their results found that miR-339-5p was most predictive for mice infection at 16 weeks, while miR-592-5p was most predictive for mice infection. When analyzing the 8-week and 16-week infection group, miR-592-5p was also most predictive of mice infection. Finally, the authors concluded that the expression levels of miR-375 and miR-200c family differed significantly during the disease process, and these miRNAs establish a link between T. forsythia and the development of periodontitis genesis, offering new insights regarding the pathobiology of this bacteria.

After reading the report, unfortunately, I found that this report was confusing for me and not clear. The study design and sample collections used in the analysis are not fully clear to me. Table 1 is confusing since, based on the detail, it shows that not all the 10 mice were infected, at the different time points, while in the M&M, the authors state that all the mice were infected.  The authors need to better describe this section.

It is not fully clear how the authors calculated the DE results at each time point i.e., from individual mouse or from the mean of multiple replicates. This obviously will affect the whole results of the analysis, interpretations, and the whole machine learning experiment.  

The authors used male and female mice; while this is very good and important to test the sex effect, they did not analyze the data based on sex. The male and female results are different, and this needs to be fully analyzed and detailed.    

The authors used a fold change of 1.1 and above was considered for analysis and considered to be significant. 1.1., which is not sufficient. This is a low cutoff.     

Minor comments:

1.     The authors need to provide more references on the miRNA i.e. their sizes. Saying small RNA is not enough and needs to state the expected size i.e. number of nucleotides.

2.     There is a need for more informative references on periodontitis on page 2, lines 69-74.

3.     The report needs English language and grammatical editing and is approved by a native English speaker.

Decision:

Finally, based on these comments, I am sorry to reject this MS, and it needs a significant revision.

Comments on the Quality of English Language

1.     The report needs English language and grammatical editing and is approved by a native English speaker.

Author Response

Comments on the IJMS MS ID 2641428 entitled ‘Unique miRomics expression profiles in Tannerella forsythia-infected mandibles during periodontitis using machine learning." written by Chairmandurai Aravindraja, Syam Jeepipalli, William Duncan, Krishna Mukesh Vekariya, Edward K. L. Chan and Lakshmyya Kesavalu

Reviewer one:

I have read the submitted research report carefully, and here I am offering my assessment and comments.

The authors studied T. forsythia, a subgingival periodontal bacterium constituting the subgingival pathogenic polymicrobial milieu during periodontitis (PD). They stated that miRNAs are pivotal in maintaining periodontal tissue homeostasis at the transcriptional, post-transcriptional, and epigenetic levels. Their study aims to characterize the global microRNAs (miRNA, miR) expression kinetics in 8- and16-weeks T. forsythia-infected C57BL/6J mouse mandibles and to identify the miRNA bacterial biomarkers of the disease process at specific time-points. The authors examined the differential expression (DE) of miRNAs in mouse mandibles (n=10) using high throughput NanoString nCounter® miRNA expression panels, which provided significant advantages over specific candidate miRNA or pathway analyses. All the T. forsythia-infected mice at two distinct time points showed bacterial colonization (100%) in the gingival surface, along with a significant increase in alveolar bone resorption (ABR) (p < 0.0001). They performed NanoString analysis of specific miRNA signatures, miRNA target pathways, and gene network analysis. They found that a total of 115 miRNAs were DE in mandible tissue during 8- and 16-week T. forsythia infection as compared with sham infection, and the majority (99) of DE miRNAs were downregulated. nCounter miRNA expression-kinetics identified 67 dysregulated miRNAs (e.g. miR-375, miR-200c, miR-200b, miR-34b-5p, miR-141) during 8-week infection, whereas 16 upregulated miRNAs (e.g. miR-1902, miR-let-7c, miR-146a) and dysregulated miRNAs (e.g. miR2135, miR-720, miR-376c) were identified during 16-weeks infection. Two miRNAs miR-375, miR- 200c were highly dysregulated with >2-fold change during 8-week infection. Six miRNAs in the 8- week infection (miR-200b, miR-141, miR-205, miR-423-3p, miR-141-3p, miR-34a-5p) and two miRNAs in the 16-week infection (miR-27a-3p, miR-15a-5p) that were dysregulated have also been reported in gingival tissue and saliva of periodontitis patients. This preclinical in vivo study identified T. forsythia-specific miRNAs (miR-let-7c, miR-210, miR-146a, miR-423-5p, miR-24, miR-218, miR-26b, miR-23a-3p) and these miRs have also been reported in the gingival tissues and saliva of periodontitis patients. In addition to DE analysis, the authors utilized the machine learning (ML) library XGBoost (eXtreme Gradient boost) to assess the impact that the number of miRNA copies has on predicting if a mouse was infected. Their results found that miR-339-5p was most predictive for mice infection at 16 weeks, while miR-592-5p was most predictive for mice infection. When analyzing the 8-week and 16-week infection group, miR-592-5p was also most predictive of mice infection. Finally, the authors concluded that the expression levels of miR-375 and miR-200c family differed significantly during the disease process, and these miRNAs establish a link between T. forsythia and the development of periodontitis genesis, offering new insights regarding the pathobiology of this bacteria.

Major comments:

  1. After reading the report, unfortunately, I found that this report was confusing for me and not clear. The study design and sample collections used in the analysis are not fully clear to me. Table 1 is confusing since, based on the detail, it shows that not all the 10 mice were infected, at the different time points, while in the M&M, the authors state that all the mice were infected.  The authors need to better describe this section.

Response: We have revised the text (M&M, results) and Table 1 for clarity. This study comprises 8 weeks and 16 weeks (two time points) of T. forsythia infection in mice. Oral plaque samples were collected at 2 weeks and  4 weeks (8 weeks’ time point), and 6 and 12 weeks (16 weeks’ time point) after infection.

  1. It is not fully clear how the authors calculated the DE results at each time point i.e., from individual mouse or from the mean of multiple replicates. This obviously will affect the whole results of the analysis, interpretations, and the whole machine learning experiment.  

Response: Two-tailed t-testing was performed on the log-transformed normalized data at each time point from individual mouse that assumed unequal variance to identify the differential gene expression (DE). The distribution of the t-statistics was calculated using the Welch–Satterthwaite equation for the degrees of freedom to estimate the 95% confidence intervals for the identified DE of miRNA between infection and sham-infection mice. We have stated in M&M statistical analysis section and we have also followed the same procedure in our prior published studies (PMID: 35563501; PMID: 36768651; PMID: 37569480).

  1. The authors used male and female mice; while this is very good and important to test the sex effect, they did not analyze the data based on sex. The male and female results are different, and this needs to be fully analyzed and detailed. 

Response: We used both males (n=5) and females (n=5) and clearly found the differences in DE between them. We have analyzed the miRNAs data based on sex and the results is presented in Table 2. We have compared female vs male (n=5) after 8 weeks of infection and found 15 upregulated miRNAs and 21 downregulated miRNAs. Similarly, we have compared female vs male (n=5) after 16 weeks of infection and found 1 upregulated and 10 downregulated miRNAs (Table 2).

  1. The authors used a fold change of 1.1 and above was considered for analysis and considered to be significant. 1.1., which is not sufficient. This is a low cutoff.  

Response: We agree with the reviewers that this is a low cut-off. Since, we studied the miRNAs from mice mandibles, the expected expression levels of miRNA are low. Further, most of our previous published study (PMID: 35563501; PMID: 36768651; PMID: 37569480). utilized the similar cut-off. Hence, in order to maintain the uniformity throughout the study, we used >1.1-fold difference with a p-value between 0.9 to 0.0001 for all our data analysis.

Minor comments:

  1. The authors need to provide more references on the miRNA i.e. their sizes. Saying small RNA is not enough and needs to state the expected size i.e. number of nucleotides.

Response: We have added miRNA size (number of nucleotides) and references in this revision.

  1. There is a need for more informative references on periodontitis on page 2, lines 69-74.

Response: We have added three references as suggested by the reviewer.   

  1. The report needs English language and grammatical editing and is approved by a native English speaker.

Response: English language and grammatical editing was completed.

Reviewer 2 Report

Comments and Suggestions for Authors

Title: Unique miRomics expression profiles in Tannerella forsythia-infected mandibles during periodontitis using machine learning.”

            In this study, the authors aimed to characterize the global microRNAs (miRNA, miR) expression kinetics in 8- and 16-weeks T. forsythia-infected C57BL/6J mouse mandibles and to identify the miRNA bacterial biomarkers of disease process at specific time-points. They concluded that the expression levels of miR-375 and miR-200c family differed significantly during disease process and these miRNAs establishes a link between T. forsythia and development of periodontitis pathogenesis, offering new insights regarding the pathobiology of this bacterium.

Minor Points:

1. Suggest rephrasing: "Consider using the term 'TNF' instead of 'TNF-α' for consistency throughout the manuscript."

2. Recommend revising for clarity: "I suggest conducting a thorough review of the text structure, as some sentences appear to lack coherence or meaning."

3. Propose a clarification: "Replace the term 'Dysregulated miRNAs' with the specific results indicating upregulation or downregulation to enhance data clarity."

4. Emphasize data normality check: "Ensure the normality of data is verified before selecting statistical methods for analysis."

5. Suggest a sentence revision: "Please review the sentence, 'This is the first monobacterial preclinical in vivo study that delineates the miRNA kinetics of T. forsythia intraoral infection.' Clarify that while the infection used only one strain (T. forsythia), the kinetics post-infection involves the restoration of the oral microbiota."

Major points:

1. Seek clarification on antibiotic treatment:

a) The antibiotic treatment (prior to T. forsythia for infected group) was also done in control group? Also, was this treatment aimed at enhancing T. forsythia colonization?

b) Is a 3-day regimen of Kanamycin plus chlorhexidine sufficient to completely suppress oral bacteria in mice? Please provide results indicating the extent of microbiota suppression following this treatment.

c) Consider highlighting in your discussion that periodontitis is often induced by dysbiosis and the accumulation of subgingival biofilms rather than the sole influence of a single human pathogen.

2. Given that T. forsythia is typically an anaerobic bacterium residing in deep gingival pockets, could you elucidate whether the presence observed on the gingival surface suggests colonization in this region, or is it possible that T. forsythia was present in the saliva? Considering the interference of saliva microbiota in the evaluation of gingival surface 16S, could you provide insights into whether T. forsythia colonized the gingival surface or if its presence is indicative of saliva contamination?

3. I recommend conducting the quantification of bone volume loss using CT scans instead of images. Given the availability of microCT data, it would be more accurate to perform volumetric measurements rather than linear measurements on images. Utilizing the microCT data will enhance the precision of your analysis.

Author Response

Reviewer Two: Comments and suggestions for Authors 

Title: “Unique miRomics expression profiles in Tannerella forsythia-infected mandibles during periodontitis using machine learning.”

            In this study, the authors aimed to characterize the global microRNAs (miRNA, miR) expression kinetics in 8- and 16-weeks T. forsythia-infected C57BL/6J mouse mandibles and to identify the miRNA bacterial biomarkers of disease process at specific time-points. They concluded that the expression levels of miR-375 and miR-200c family differed significantly during disease process and these miRNAs establishes a link between T. forsythia and development of periodontitis pathogenesis, offering new insights regarding the pathobiology of this bacterium.

Minor Points:

  1. Suggest rephrasing: "Consider using the term 'TNF' instead of 'TNF-α' for consistency throughout the manuscript."

Response: We have rephrased as suggested.

  1. Recommend revising for clarity: "I suggest conducting a thorough review of the text structure, as some sentences appear to lack coherence or meaning."

Response: We have revised the text for clarity.

  1. Propose a clarification: "Replace the term 'Dysregulated miRNAs' with the specific results indicating upregulation or downregulation to enhance data clarity."

Response: Replaced the suggested word throughout the manuscript.  

  1. Emphasize data normality check: "Ensure the normality of data is verified before selecting statistical methods for analysis."

Response: NanoString miRNAs data were analyzed and normalized using nSolver™ 4.0. Software Analysis (NanoString Technologies, Seattle, WA, USA). We have stated in the M&M section.

  1. Suggest a sentence revision: "Please review the sentence, 'This is the first monobacterial preclinical in vivo study that delineates the miRNA kinetics of T. forsythia intraoral infection.' Clarify that while the infection used only one strain (T. forsythia), the kinetics post-infection involves the restoration of the oral microbiota."

Response: We have revised the sentence for clarity in this revision.

Major comments:

  1. Seek clarification on antibiotic treatment:
  2. a) The antibiotic treatment (prior to forsythia for infected group) was also done in control group? Also, was this treatment aimed at enhancing T. forsythia colonization?

Response: The antibiotic treatment was done in both T. forsythia infected and control groups. The reviewer is correct that antibiotic treatment aimed at enhancing T. forsythia colonization.

  1. b) Is a 3-day regimen of Kanamycin plus chlorhexidine sufficient to completely suppress oral bacteria in mice? Please provide results indicating the extent of microbiota suppression following this treatment.

Response: Kanamycin plus chlorhexidine treatment for 3 days will not be sufficient to completely suppress oral bacteria in mice but will reduce oral microflora that will facilitate T. forsythia colonization/infection. Antibiotic pretreatment of mice and rats are standard procedure in rodent studies with oral bacteria for the last 3-4 decades. There are several published studies in the literature using antibiotic pretreatment of mice and rats and we have published several studies. We have not examined (Illumina 16S rRNA sequencing) mice oral flora before and after antibiotic pretreatment.    

  1. c) Consider highlighting in your discussion that periodontitis is often induced by dysbiosis and the accumulation of subgingival biofilms rather than the sole influence of a single human pathogen.

Response: We highlighted that periodontitis is often induced by dysbiosis and the accumulation of subgingival biofilms rather than the sole influence of a single human pathogen.  

  1. Given that T. forsythia is typically an anaerobic bacterium residing in deep gingival pockets, could you elucidate whether the presence observed on the gingival surface suggests colonization in this region, or is it possible that T. forsythia was present in the saliva? Considering the interference of saliva microbiota in the evaluation of gingival surface 16S, could you provide insights into whether T. forsythia colonized the gingival surface or if its presence is indicative of saliva contamination?

Response: After 3 days following T. forsythia infection (4 days infection), T. forsythia (genomic DNA) presence on the gingival surface suggests its colonization/infection. It is possible that T. forsythia may be present in the saliva (To best of my knowledge, there is no free-flowing saliva in mice). Our prior published studies using fluorescent in-situ hybridization (FISH), demonstrated the viable periodontal bacteria in the gingival tissue, aorta, synovial membrane, salivary glands, and brain. FISH will be done in gingival tissues to quantify 16S rRNA, which is present only in bacteria that are actively synthesizing proteins. Human oral bacteria are commensal and species-specific; therefore, these cannot normally colonize the mouse oral cavity for long periods of time. Hence, multiple gingival infections over a long period facilitate human bacterial colonization in the mouse oral cavity. As human specific oral bacteria are difficult to colonize/infect (epithelial attachment, invasion, multiplication) in mice oral cavity, we perform alternate week infection cycle to monitor bacterial adhesion/colonization. In future, we will perform oral plaque sampling without saliva contamination by swabbing the mouth with sterile gauge before sampling. In the past, several studies have isolated the bacteria after infection by culturing it in an appropriate culture medium.    

  1. I recommend conducting the quantification of bone volume loss using CT scans instead of images. Given the availability of microCT data, it would be more accurate to perform volumetric measurements rather than linear measurements on images. Utilizing the microCT data will enhance the precision of your analysis.

Response: We agree with the reviewer that microCT quantification of alveolar bone volume loss will be more accurate to perform 3-dimensional volumetric measurements rather than 2-dimensional linear measurements on images. We will plan to do microCT analysis in future studies.

Round 2

Reviewer 2 Report

Comments and Suggestions for Authors

I would like to thank authors for the response, but there are some changes needed:

-  Regarding the sentence on the abstract for better comprehension: Please consider revising the sentence concerning the abstract to enhance clarity. Specifically, the phrase 'both the 8 weeks and when the 8-week and 16-week infection Nanostring results were grouped together...' could benefit from clarification.

- Concerning the sentence on Kanamycin administration: As the authors did not conduct 16S sequencing of the oral microbiota, the statement 'Kanamycin (500 mg/ml) was administered to all mice in sterile drinking water for three days to suppress the mouse oral bacteria followed by rinsing with 0.12% chlorhexidine gluconate...' should be modified. The original intent was not to suppress the oral microbiota, as clarified by the authors.

- Regarding the mandible pictures: Since the authors did not perform microCT analysis, improvements are needed for the pictures displaying mandibules and indicating alveolar bone loss. Please include a scale bar, ensure uniform picture sizes, and verify that the numbers denoting the area are adequately visible as they appear too small in the current images.

  1.  

Author Response

REVIEWER 2: Comments and Suggestions for Authors

I would like to thank authors for the response, but there are some changes needed:

Comments:

  1. Regarding the sentence on the abstract for better comprehension: Please consider revising the sentence concerning the abstract to enhance clarity. Specifically, the phrase 'both the 8 weeks and when the 8-week and 16-week infection Nanostring results were grouped together...' could benefit from clarification.

Response: We have revised the sentence for clarity. XGBoost found that miR-339-5p was most predictive for mice infection at 16 weeks. miR-592-5p was most predictive for mice infection at 8 weeks and also when the 8 week and 16-week results were grouped together.

  1. Concerning the sentence on Kanamycin administration: As the authors did not conduct 16S sequencing of the oral microbiota, the statement 'Kanamycin (500 mg/ml) was administered to all mice in sterile drinking water for three daysto suppress the mouse oral bacteria followed by rinsing with 12% chlorhexidine gluconate...' should be modified. The original intent was not to suppress the oral microbiota, as clarified by the authors.

Response: There is some confusion. The purpose (original intent) for Kanamycin administration and 0.12% chlorhexidine gluconate rinse was to suppress mouse oral microflora. This suppression will facilitate human bacteria colonization/infection. 

  1. Regarding the mandible pictures: Since the authors did not perform microCT analysis, improvements are needed for the pictures displaying mandibles and indicating alveolar bone loss. Please include a scale bar, ensure uniform picture sizes, and verify that the numbers denoting the area are adequately visible as they appear too small in the current images.

Response: We have made changes in the Figure 1 and the numbers denoting the alveolar bone resorption area are now clearly visible. While capturing mouse images for measurement, we have taken as much care to ensure uniform sizes. To best of our knowledge, the software that displays images do not have scale bar embedded in it.